# How to Tailor Educational Maze Games: The Student's Preferences

Valentina Terzieva [1], Boyan Bontchev [2,*], Yavor Dankov [2] and Elena Paunova-Hubenova [1]

1   Institute of Information and Communication Technologies, Bulgarian Academy of Sciences, Acad. G. Bonchev St., Bl. 2, 1113 Sofia, Bulgaria; valentina.terzieva@iict.bas.bg (V.T.); elena.paunova@iict.bas.bg (E.P.-H.)
2   Faculty of Mathematics and Informatics, Sofia University, "St. Kl. Ohridski", 1164 Sofia, Bulgaria; yavor.dankov@fmi.uni-sofia.bg
*   Correspondence: bbontchev@fmi.uni-sofia.bg

**Abstract:** Personalized learning has gained in popularity over the past decade. It provides learners with learning resources that comply with their characteristics and preferences or offers them tasks and quizzes adapted to their performance. This research presents how we apply this concept to an educational video maze game created and generated on the APOGEE platform. In particular, this article explores the following three research questions: (1) Which characteristics in the student's model should be considered for the personalization of educational video games? (2) What are the student's preferences regarding the personalization of educational video games? (3) How should the process of personalization of educational video games be organized? The answers to these questions are found by conducting practical experiments concerning user experience with the educational maze video game. The article also describes the model of students comprising user's, learner's, and player's aspects with both static and dynamic features. Further, the personalization process of educational games based on this model is described. The results showing the student's preferences are presented and critically examined. The provided discussion involves the disparities in the preferences of different groups of students concerning the amounts of play of learning games, preferred mini-games, and parameters to which educational materials should be tailored.

**Keywords:** personalization; game-based learning; educational games; maze video games; APOGEE

## 1. Introduction

The recent integration of digital technologies in all areas of society, known as digital transformation, has a considerable impact on all organizational, economic, and social activities. Many new models and approaches have appeared, especially in the educational area. The term *serious game* as a type of game intended for learning purposes was first introduced in [1]. Since then, the definition of the term has evolved, and still there is no single one established. Such games are described as games with at least one purpose other than pure entertainment [2,3]. Educational games are considered a part of serious games used in various educational areas: from early childhood to university, in vocational and special education, etc. [3]. Recently, many educators have begun to employ game-based learning (GBL) in various subject areas to improve student motivation, facilitate knowledge acquisition, and develop problem-solving skills [4]. Using innovative visualization technologies (e.g., laser projections) may increase the involvement of the students in game-based learning and game attractiveness [5]. Serious games—games created with a specific educational purpose in mind rather than for entertainment—are those that help achieve these goals [6]. The outcomes of integrating the games in an educational context depend on the applied instructional framework and the availability of technological infrastructure enabling the performing of gaming activities [7,8]. The efficiency of educational games as a pedagogical tool is strongly related to the built-in learning basis of in-game activities. The motivation for the game implementation in the educational process is to achieve a specific outcome (learning enhancement) by changing student attitudes to learning to a positive one [9].

Serious games can be treated as methods and tools to improve the understanding and interpretation of various concepts, so they are appropriate for sustainable education. The attractiveness of video games as a pedagogical approach lies in the fact that they are immersive and allow the students to acquire knowledge of everything that happens during the play, which leads to a positive change in attitude, as well as in behavior, which in fact induces sustainable outcomes [10]. Moreover, results usually indicate enhanced student motivation when using game-based learning compared to the traditional process. It is essential, especially in sustainability education, to provide multidisciplinary and interdisciplinary learning spaces to create opportunities for exploring and assessing a variety of assumptions to ensure a more efficient interpretation of the complicated issues of sustainability. There is a need for new paradigms allowing us to understand the complexity of sustainability and its relationship to other knowledge. Serious games are among tools that recently appear to be especially suitable for this purpose [11].

Serious games help adolescents realize the significance of various topics that may be neglected if they are not demonstratively presented. Didactic games increase learners' concern for school subjects [7] and crucial issues such as preserving culture and historical heritage, climate changes [12], healthy lifestyle [13], social causes [14], etc. In addition, serious games can address three main learning goals concerning sustainability issues: (i) introducing players to some of the crucial challenges related to the area of sustainability; (ii) providing general knowledge and ideas on the sustainability topic; and (iii) encouraging players to develop and propose solutions or take actions that contribute to the sustainability of a particular environmental or socioeconomic field [10]. These three main learning goals have been addressed in an educational video game related to the sustainability of monumental cultural heritage [12] created by the authors through the application of the APOGEE platform [15].

Personalized learning has gained in popularity over the past decade. In the context of technology-enhanced learning, personalization usually concerns customizing digital learning resources to the needs and preferences of a particular learner in addition to individual feedback [16]. It means not using a "one-size-fits-all" approach but providing learners with learning resources that comply with their age, prior knowledge, preferences, etc., or adapting tasks and quizzes to their performance [17]. At first glance, the idea of personalization is appealing, though its implementation is complex.

Three significant issues regarding the process of tailoring educational video games form the research questions explored in this article:

1. Which characteristics in the student's model should be considered to personalize educational video games?
2. What are the student preferences regarding the personalization of educational video games?
3. How should the process of personalization of educational video games be organized?

This study presents some of the findings from the conducted practical experiment with the educational video maze game, created and generated on the APOGEE platform [15] and played by students. The users (before starting to play the game) fill out particular questionnaires to collect information about their static and dynamic characteristics. The findings of the online questionnaires are used to create the student's model, used for the process of personalization of maze video games on the platform. The practical experiment results are described in this paper and show the students' results and preferences.

The article is structured as follows. The study starts with an introduction and a brief outline of related works concerning educational video games, personalized game-based learning, and learner modeling. In Section 3, the student-centered personalization of educational maze games in APOGEE is presented, including the APOGEE personalization framework, the student's model used in the APOGEE game platform, and the process of personalization. Section 4 presents the survey results. Section 5 gives a discussion and limitations of the study. The article ends with a conclusion and future works.

## 2. Related Works

### 2.1. Educational Video Games

With the development and fast pace of information and communication technologies' (ICT) entry into the learning process [18], educational video games are gaining more and more popularity and wide application in different areas. They are a variety of serious video games with applications in different areas such as health, defense, education, and many others. Educational video games allow learners to study by playing the game (game-based learning), as well as supporting a high level of motivation and engagement [19], and also to motivate students and enhance their learning process [20]. Educational video games are a technological innovation [20]. These games are increasingly integrated into education and e-learning systems and environments [21,22]. Educational video games consist of various content, such as didactic and game content, game elements, digitized art, and artifacts [23], representing essential aspects of the game-based learning process [4]. All these aspects are combined into a well-created virtual design [24] that seeks to meet all the requirements of the users and digital environment [25]. Therefore, this makes the game more fascinating to players and learners, contributing to the overall user experience of the game. The recommendation and selection of didactic content to integrate into these games are intended to be according to the target group of learners and represent a challenge in technology-enhanced learning [26]. The designers of educational video games should carefully design the video games for learning to successfully fulfill the defined educational tasks and achieve the set goals. In this regard, the design and creation of educational video games are supported by various means such as:

- Specialized instruments that assist designers in realizing this process [27] by facilitating their work and providing the opportunity to use them for analysis and evaluation of the designed games [28].
- Implementing personalization of the learning content and adaptation of gameplay of the games depending on the various characteristics and behaviors of players/learners, which results in improving the design of the educational video games and improving the user experience of players/learners in games [29].
- Specialized platforms for creating educational video games to support and facilitate design processes and create educational video games, and within these platforms all these processes are realized [30].

Achieving a successful personalization and adaptation in educational video games is a significant factor in reaching educational goals and meeting the requirements of the defined educational group from learners for which the game is designed and created. Considering the general aspects of design in a virtual environment and specifics [24,31], a well-designed, personalized, and adapted video game provides an engaging and authentic environment [32] for a specific educational group of learners. Thus, it provides opportunities for easier and faster perception and learning of educational content in the game and an improved user experience for learners and players.

### 2.2. Personalized Game-Based Learning

Recently, various research reports that personalized learning enabled by technology has been identified as an emerging trend that can significantly influence teaching and learning [33,34]. Much research has already proved the greater efficiency and usability of learning systems customized to users compared to usual ones. Their main benefit is the significant improvement in the learning performance and outcomes [35,36]. Such systems utilize different models of users and a concept for reshaping and reusing learning resources to achieve a personalized learning experience. The structure and features of learning content presented in learning resources, together with the availability of rich metadata, are essential for providing functionalities for resource customization to different users.

Personalization in a learning context can be considered a complex, multifaceted term with different meanings depending on the viewpoints of key users: students, teachers, or policymakers [37]. Thus, personalization embraces dimensions such as: (i) what to learn

(the learning content and curriculum); (ii) why to learn a particular thing (the learning goals); (iii) how to learn (the learning approach and learning path); (iv) when to learn (the learning pace); (v) where the learning process takes place (the learning context); (vi) who is learning (the individual or particular group of students). On the other hand, personalization in a learning context can occur at a macro-level (e.g., the educational goal, approach to teaching and learning, etc.) or at a micro-level (e.g., the context, the pace, and the learning paths). All the issues mentioned above also apply to serious games for learning.

The research specified the main factors that have to be reflected in the process of game development to ensure the effectiveness of game-based learning: (i) goal-oriented structure; (ii) reinforcement of engagement and performance by stimulating rewards; (iii) easy tracking of student achievement; (iv) a game environment that supports students' competitive behavior; and (v) fun and enjoyment gameplay [38]. We propose two additional factors: (vi) personalization of game content and (vii) adaptation of gameplay.

Adaptation and personalization are vast research topics in serious games, especially in the educational area. Generally, adaptation is considered a process of adapting, i.e., something is changed or modified to become suitable for new conditions, needs, or situations [39]. Adaptive learning systems aim to provide personalized applications and services by using adaptive web technologies, thus enabling traditional web-based educational systems to be more beneficial to individual learners [40,41]. Such adaptive educational systems usually gather data about the students' interactions within the system and create models of students. Then, the system utilizes these data and models to adapt the presentation, navigation, and annotation of learning content, before providing it to the students [42].

In the game context, the term personalization usually refers to the customization of some of the game aspects (features, interface, content, etc.) to the needs, preferences, and other characteristics of a user. However, the adaptation typically relates to the constant modification of the gameplay considering the user's in-game interactions and performance [43]. Usually, the terms personalization, adaptation, customization, and tailoring have similar meanings and contexts. The dimensions, principles, and methods used in adapting serious games can be various, e.g., adaptive levels and content generation, interactive and adaptive storytelling, adaptive content sequencing and presentation, adaptive guidance, feedback and hints, intelligent supporting agents, etc. [44]. For educational games to be customizable, the built-in learning content has to be well designed and structured in relatively small learning resources.

### 2.3. Modeling of Student Features

The approach to the personalization of educational services is based on the model of students, which follows general user modeling techniques. There are three main issues to consider in the user modeling process: dimensions that matter, model representation, and manners of obtaining user information [45]. The motivation for adaptation and personalization is based on the view that variety in some characteristics of users can affect the usability and efficiency of the services provided to them. Hence, it is believed that if the functionality of a system is tailored according to users' characteristics, it will provide increased benefits to users. Dimensions that matter are essential for the adaptive systems based on modeling students [42].

Student models provide information and evaluations on the learners' mastered knowledge and skills, competencies and achievements, learning goals and preferences, cognitive and emotional skills, etc. [46]. User modeling also supports tracking how and when each student learns skills, what pedagogical approaches give the best results, and even reflects students' cultural preferences and individual interests [47]. For adaptive educational services, it is essential to be familiar with some basic demographic characteristics of a user, e.g., name, age, gender, and grade. Such information presents static parameters that can also count in educational settings.

In an educational context, students' knowledge and background are the main characteristics affecting the adaptability of a system. One of the traditional approaches to

modeling student knowledge is mapping with a granular conceptual structure of a particular learning domain, so this characteristic is dynamic (it changes over time). Usually, background concerns relevant experience before using the system, so this parameter is typically static and stereotypical. Most often, it is represented with several predefined values, e.g., beginner, advanced, and expert [48,49].

Users' interests and preferences are also considered when applying adaptive teaching. They reflect the presentation of educational content and are closely related to learning styles. Intelligent dialog systems usually use some kinds of modeling of users' goals and needs. Such an approach is prevalent in many web-based adaptive systems, especially adaptive recommender systems [50]. Within adaptive educational systems, the student's goal serves to propose the optimal strategy for efficient learning, thus achieving the ultimate student's objective, to learn and comprehend the educational content.

Recently, with the development of information technologies that obtain people's emotions, they often are considered an additional source of information for tailoring adaptive systems to users. Much research has found that emotions influence learning, especially attention, memory retention, reasoning, problem solving, and recall [51,52]. The second significant aspect of user modeling concerns how the selected user characteristics are modeled. Usually, most of the used approaches are based on stereotypical and overlay user modeling [46,53]:

- Stereotype user modeling: The idea is that if users use some systems similarly, usually they have similar sets of features, so they can be grouped into categories called stereotypes. A user's description involves a single or a combination of several predefined stereotypes. This approach has the advantage of deriving sufficient information for modeling from little evidence. However, precise overlay models should be used for modeling fine-grained users' characteristics.
- Overlay user modeling: This approach usually is used for modeling student knowledge and skills as a subset of subject domain knowledge, which is ingrained into components (e.g., topics, concepts, knowledge elements, and outcomes), representing pieces of declarative domain knowledge. The main benefit of this model is the flexibility, precision, and the ability dynamically to reflect the development of users' characteristics. However, there is still a need to build a precise formal domain model, which can be a difficult task.

Usually, adaptive systems acquire information for user modeling in two manners [45]: (i) by asking users to directly input some data and features or (ii) by extracting it, taking into account user interactions in the system (i.e., using machine learning and artificial intelligence techniques exploiting the logs files). Often, systems utilize some combination of these approaches and additionally use user feedback derived from intelligent dialog systems. Another approach is the active involvement of users in the modeling process by enabling them to modify their user models [54].

## 3. Student-Centered Personalization of Educational Maze Games in APOGEE

For an effective and efficient student-centered tailoring of educational maze games, the APOGEE project [15] started the development of a software platform for an automatized generation of customizable 3D video maze games based on their formal description [30] and the practical validation of the platform by online experiments with constructed game prototypes in the context of Bulgarian medieval history.

The educational maze games generated within the APOGEE software platform can be customized by applying the personalization framework presented in Figure 1. This framework outlines the three key groups of drivers of the process of personalization and adaptation:

- Demographics, preferences, and goals of the students, which are self-reported in a questionnaire with predefined values.
- Learning and playing styles derived from students' answers on specially designed online questionnaires.

- In-game measured performance and efficiency, calculated using parameters measured during gameplay.

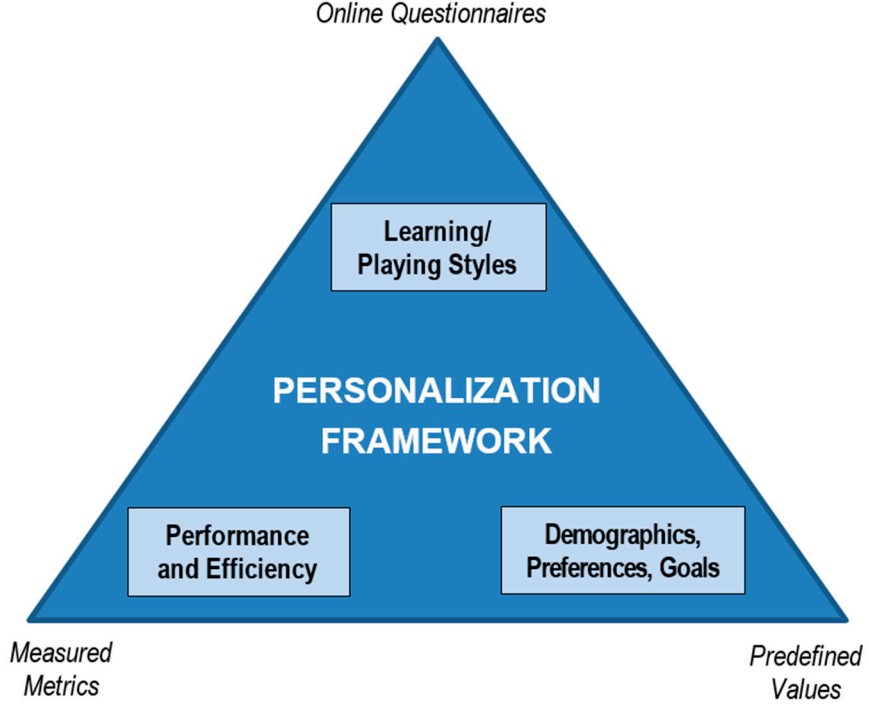

**Figure 1.** APOGEE personalization framework.

Demographics, preferences, and goals of each student, together with his/her learning and playing styles, are static features suitable for personalization of his/her game instance before the game session. In such a manner, the personalization framework gathers data necessary to create customized instances of video maze games that suit the characteristics and preferences of each student.

On the other hand, the performance reflects the obtained outcomes from the mini-games, while the efficiency is presented by the ratio between the performed effect and the effort (or time) required to achieve it. Both the performance and efficiency are applied for dynamic adjustment of some game features while playing the game, such as task difficulty, NPC (non-player character) behavior, and audio-visual effects.

### 3.1. Student Model

The developed model of a student comprises three main aspects: user, learner, and player, where each aspect is presented with both static and dynamic parameters (Figure 2). A detailed explanation of this model is given in [29,55].

Here, we make a brief introduction to the parameters of the student's model, outlining the most significant characteristics of each aspect:

- User: *static features* comprise general demographic and user data (e.g., age, gender, and identification), while *dynamic features* reflect an aggregated score of the game and gathered data about the emotional and arousal state of the user during gameplay.
- Learner: *static features* concern specific data necessary for personalization purposes (e.g., knowledge level, learning goals, and learning style), while *dynamic features* represent in-game achievements when solving built-in mini-games (e.g., acquired knowledge, score, and efficacy).
- Player: *The static features* regard playing style and playing goals used for gameplay tailoring. *Dynamic features* (e.g., collected objects, gained points, speed, and efficiency) are the base for dynamic difficulty customization of particular gameplay parameters.

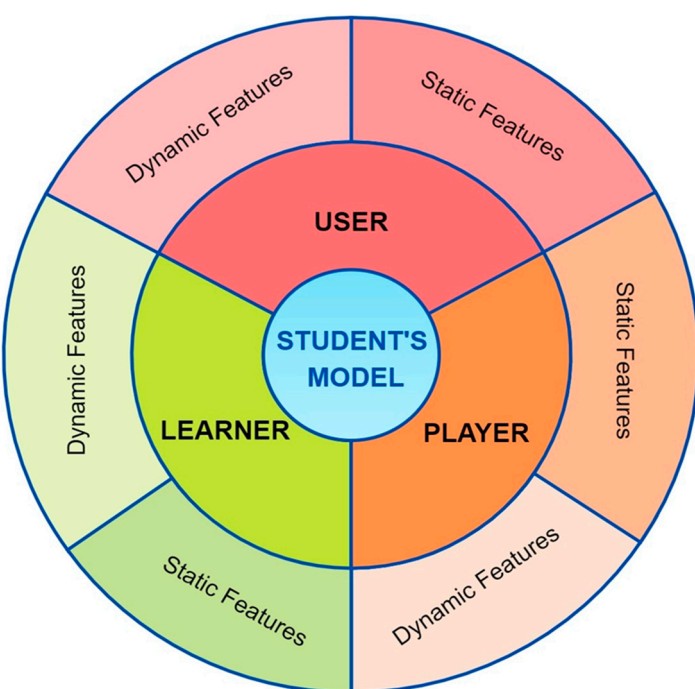

**Figure 2.** Student's model used in the APOGEE game platform.

For each registered student, the game achievements in each game session are stored in a log file in the database. Based on them, the processing and analytical tools of the APOGEE system can calculate various statistical parameters concerning learning and gameplay processes. These are (i) the in-game efficiency, time, and score for the solved mini-games (puzzles) and various correlations with student's characteristics (age, knowledge level, playing and learning styles, etc.) and (ii) the effect of the maze game customization, including personalization of learning content in mini-games and dynamic difficulty adaptation of gameplay. Further, many other statistics and different correlations between static and dynamic parameters of the student's model can be calculated to reveal the hidden dependencies and relations.

The *static features* in the student model are obtained by direct input from the students when filling in a questionnaire with stereotype categories and open answers. Static features describe relatively unchangeable over time or slowly changing student characteristics, such as demographic data, the initial knowledge level in the learning domain, learning and playing styles, and the learning and playing goals. They serve for user identification and initial setting up, personalization, and choice of the preferred type of the mini-games and learning content presented in the maze. The *dynamic features* indicate attributes that vary over time and reflect learning performance and efficiency during learning by playing sessions. The indicators for these features are tracked for each game session and calculated by the APOGEE platform. They reflect outcomes during play sessions (e.g., mini-games played, points gained, objects collected, playing speed, and knowledge acquired). A pre-game questionnaire is used to infer the learning and playing styles of students following psychological theories [56].

We are going to apply this student's model for tailoring both the entire educational maze game and the mini-games embedded in the maze. Static features in the models will be used to personalize the whole maze (i.e., the structure, choice, and distribution of mini-games, educational content shown on learning boards and in learning tasks and level of difficulty) and each instance of a mini-game. On the other hand, dynamic features such as individual in-game performance and emotional state will help dynamical adjustment of task difficulty, game flow, mini-game feedback (help), and NPC behavior [57,58].

### 3.2. The Process of Personalization of Maze Video Games

Our approach to the personalization of educational maze games is based on the model of students, which follows general user modeling techniques. There are three main issues to consider in the user modeling process: dimensions that matter, model representation, and manners of obtaining user information [45]. The motivation for adaptation and personalization is based on the understanding that changes in some students' characteristics can affect the usability and efficiency of the services provided to them. For this reason, we focus on the users and their characteristics (described as static and dynamic features) utilized in creating the student's model used in the APOGEE game platform. This model can be updated according to the users and changed and created for new users of the platform. We argue that this will benefit the process of personalization of educational maze video games on the APOGEE platform. Hence, it is believed that if the functionality of a system is tailored according to users' characteristics, it will provide increased benefits to users.

When starting a game, the system asks the student to register and fill out two questionnaires to derive the data needed for creating the student's model (Figure 3). All attributes describing both static and dynamic parameters of the three main components of the student's model (user, learner, and player) are initialized (Figure 2). Thus, each student has to select one of the predefined categories of static parameters (e.g., school age, gender, initial knowledge level, and learning goal) depending on self-reported data. For example, initial knowledge is self-assessed before the game session, at which point the student has to choose one of several stereotypes such as beginner, advanced, and expert. For learning goals, the student can opt between an introduction to the subject, a game with experiments, a detailed study, an assessment game, and a summarization. In addition, the student's preferred learning and playing styles are determined. Then, this initial model of a student is updated continuously in the gameplay process. Dynamic parameters in the three main components of the model are amended by gathering and processing current values of the data concerning in-game interactions and performance in the game-based learning process. The selection of both the learning content and mini-games should depend on student age and learning style. Since the curriculum of any learning subject is decided by factors outside the particular student, it is not based on gender and playing style, so the selection of learning content does not depend on these two characteristics. Teachers determine it according to students' age, knowledge level, and learning goals and style. However, the selection of mini-games depends neither on the knowledge level of individual students nor on their learning goals. Survey results below confirm that mini-games should be selected according to student gender and playing style, together with age and learning style.

The personalization framework within the APOGEE platform enables the generation of various instances of an educational video maze game according to the model of each student. This maze game consists of several halls, which display content concerning some topic. In each hall, there are up to eight (two on each of the four walls) information boards for narrative and visual content delivery and an instance of several types of mini-games suitable for learning-by-doing activities. These boards have multiple slides presenting learning objects such as text, images, and diagrams that provide the necessary knowledge for performing didactical tasks in mini-games in the same hall. The type of built-in didactic mini-games and their parameters (e.g., level of difficulty, the type, volume, and presentation of educational content and assigned points) can be chosen in such a manner to comply with the characteristics of the students reflected in their models. In addition, the same learning content may have representation in several different mini-games (puzzles).

Further, hints and feedback concerning each mini-game can be pre-selected or changed dynamically depending on the teacher's decision and the student's profile. Thus, the educational maze game can be generated in different versions with various built-in mini-games and educational content to be personalized and suit students with diverse characteristics and preferences.

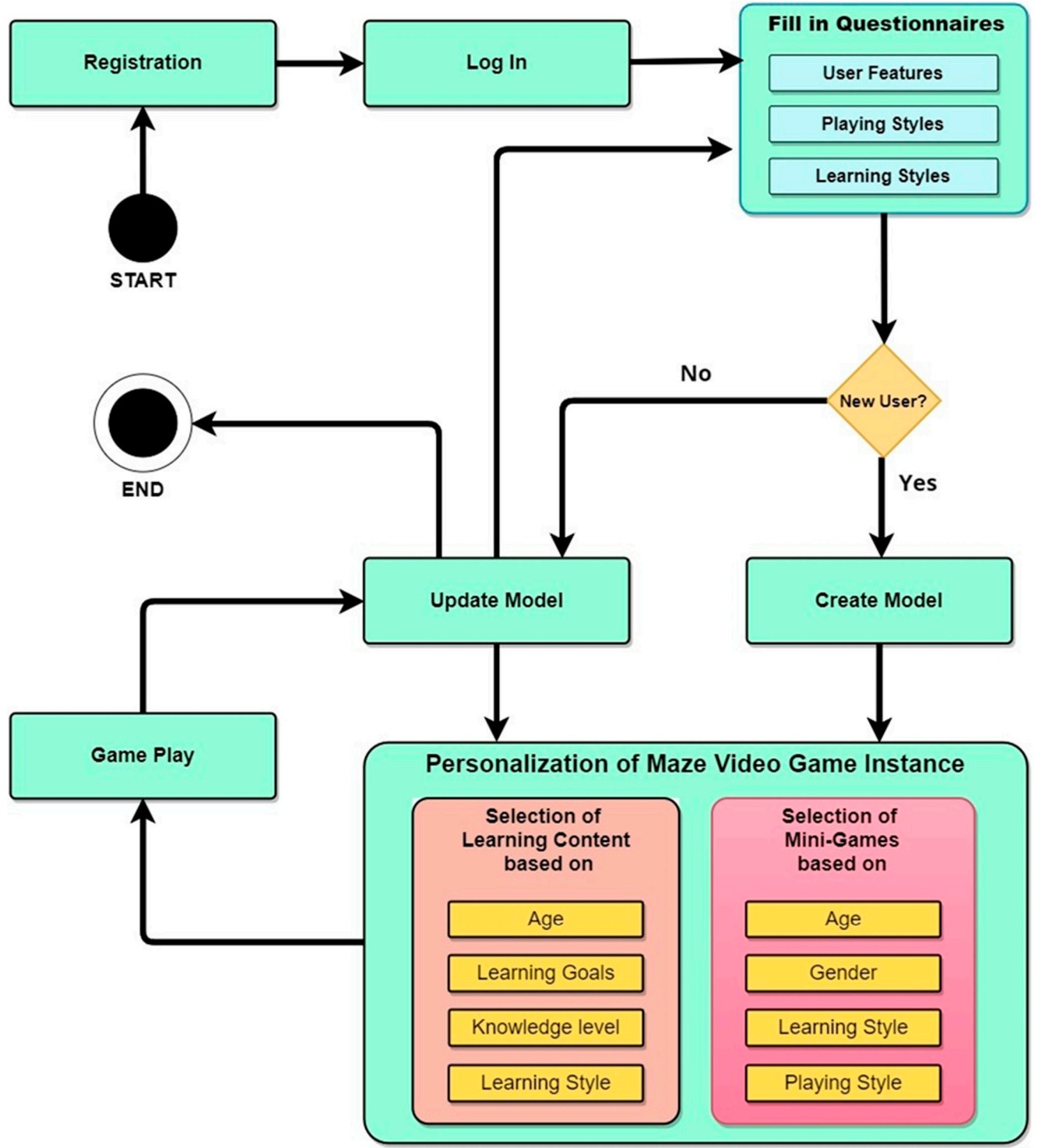

**Figure 3.** The process of personalization of educational maze video games in the APOGEE platform.

### 3.3. Student-Centered Personalization of Educational Mini-Games

The main game represents a maze that plays the role of a container for puzzle mini-games introducing the educational tasks. The topic of the mini-games and the information boards in one hall corresponds to the learning material of one lesson. The demonstration game is in the domain of the medieval history of Bulgaria. The player's goal in the main game is to pass through all the halls, completing the mandatory tasks in the form of puzzle games by collecting the maximum score. To gain extra points, knowledge, and fun,

participants have the opportunity to play mini-games in the maze, which are optional. The non-mandatory mini-games provide the teacher who sets them up with the opportunity to achieve other pedagogical goals tailored to students' specific play and learning styles. Students can focus on topics that are interesting to them or they need to learn more facts about. The different types of puzzle games are grouped according to the actions required by the player to pass them successfully and are presented below:

- **Question games**: Answering short questions with open or closed answers. The complexity of the questions, the number of suggested answers, and the hints can be customized. The following games are included in this group:
    - Question to unlock the door to another room ("Open Sesame!") (see Figure 4a). When approaching the door, the question appears with predefined answers, from which the player has to choose the correct one. Depending on the learner's knowledge level, the difficulty and content of the question are pre-set.
    - A quiz of several consecutive questions for passing to another hall, at several levels, with various types of questions for testing the knowledge acquired in the game. In this mini-game, the difficulty of the questions and the passing threshold can be personalized.
- **Games for searching or matching objects**: Aiming at finding game objects according to specific criteria. Personalization can be performed by changing the number and size of the searched object or word. This group includes the following puzzle games:
    - Search for translucent objects in the maze halls ("I see you") (see Figure 4b). The type of objects is related to the learning matter in the hall, and the objects found can be used in other mini-games. Depending on the player's characteristics, the objects' size, type, and number can be changed.
    - Search for hidden objects covered by larger ones ("Find me!"). Analogically to the previous game, the objects are related to the learning material and can be used in subsequent puzzles. Here, the objects' size, type, and number can differ.
    - Search for pairs of identical or matching cards in multiple gridded cards ("Memory"). The game develops memory and helps the player to remember analogies. The number of cards and the matching criterion (two identical images; image and description; or related objects) can be personalized in this game.
    - Word search in a grid of letters ("Word soup") (see Figure 4c). Searched words are terms from the learning material that can be positioned in different directions. The grid size, the words' length, and their orientation (horizontally, vertically, or by diagonals) can be changed according to the age and knowledge of the player.
- **Sorting or classification games:** Games for sorting or arranging different objects, including puzzles and letters. The number and content of the arranged items can be customized. This group contains the following mini-games:
    - Sorting objects into groups according to a given feature ("Divide & Conquer"). The objects involved in this mini-game can be obtained after playing a game from the previous group or obtained specifically for grouping. The sorting criteria, the number, and the type of objects can be personalized according to the difficulty required and the student's age and gender.
    - Put in order a didactic image, a "2D puzzle" related to the learning matter. In this mini-game, personalization is carried out by image outlines, turning, and dimensions of the puzzle pieces.
- **Action games**: The user needs to interact with the game objects to finish the mini-game successfully. The object number and possible choices can be personalized. The educational maze contains two action games:
    - Shooting at inanimate objects moving in the hall (first-person shooter, or FPS). The player must hit flying balloons with attached objects (artifacts) related to the learning material. The obtained artifacts can be used in other mini-games, such

as the above-described one (separating objects into groups). Personalization is carried out by changing the balloon's volume, velocity, range, and direction of movement.

- Rolling balls on the floor ("Roll a ball") (see Figure 4d). The player has to move a ball to a target area on an image or a given object (e.g., a ring), matching the label above them. The personalization can be performed by the number of balls and their possible end positions.

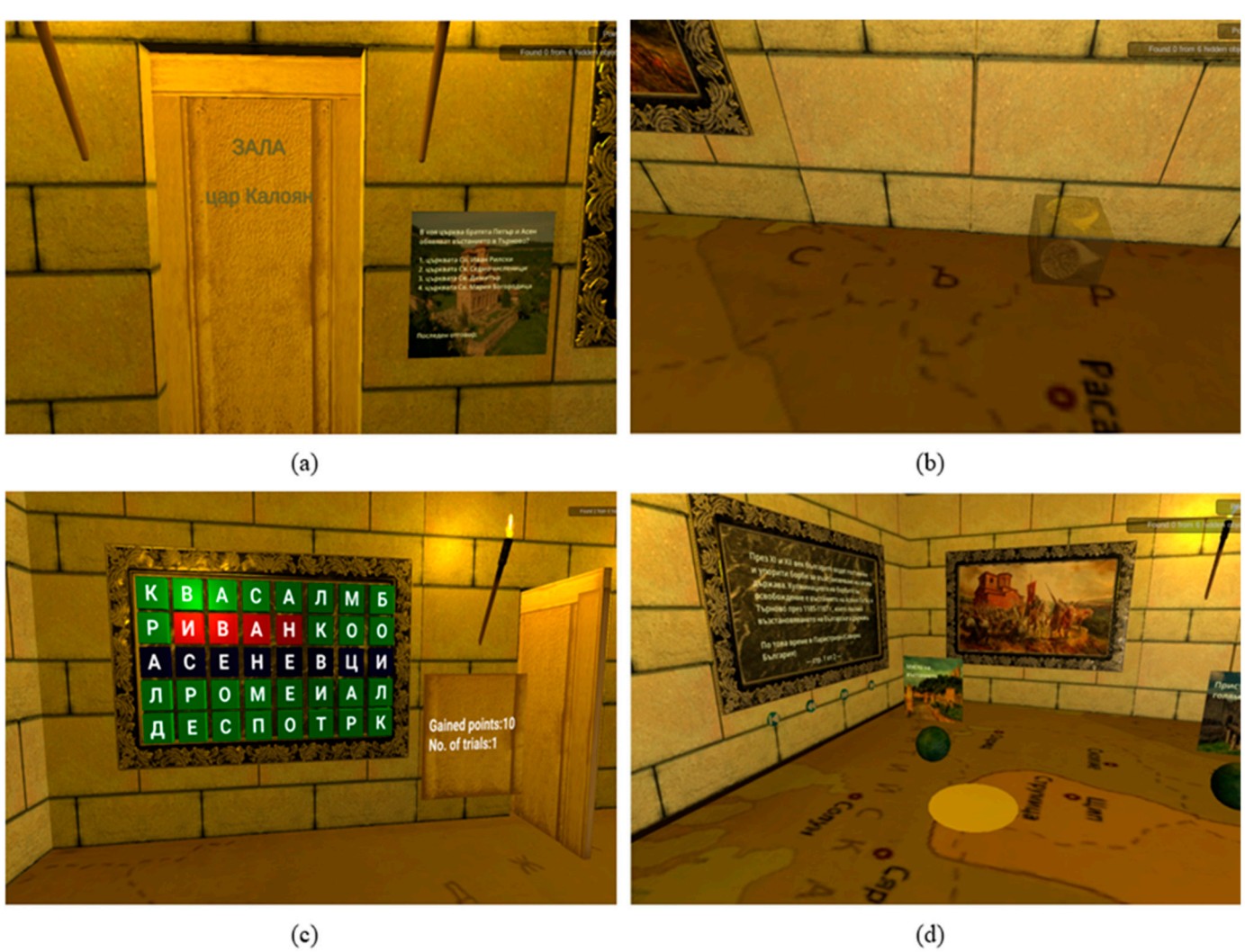

**Figure 4.** Screenshots of the mini-games "Open Sesame!" (**a**), "I see you" (**b**), "Word soup" (**c**), and "Roll a ball" (**d**).

Personalization of the main game to the students' specific characteristics is achieved by tailoring the mini-games' parameters to the players. The most frequently modified mini-game parameters are the embedded learning content, the complexity, and the game objects used.

## 4. Survey Results

The results presented in this section are obtained by following a methodology for conducting an online survey after watching a demonstration video record of gameplay and an online maze game session with various mini-games included in maze halls.

### 4.1. Methodology

The authors created a questionnaire (see the Appendix A) focused on students' attitudes towards the various puzzle games included in the larger maze game container. Another aspect explored is the possible options for customizing them according to the learning and playing styles of the respondents. Participants were invited by the local web site of the European Researchers' Night in Bulgaria and instructed to follow the demonstration, play the online game, and fill in the online survey. In the frame of the events held in 2020 and 2021, the game content was introduced to K12 and university students through a video clip of a few minutes in duration, presenting the playing of the main game with a historical focus. In the halls, a variety of puzzle games (see Section 3.2) are available, including searching and arranging objects and asking questions with predefined answers. After watching the video, participants had the opportunity to play the game so they could adequately appreciate the personalization of the educational games. To differentiate playing preferences according to student age, both university and K12 students played the same maze game. Then, they responded to the same questions about their preferences for mini-games included in the maze. All the respondents completed the online survey voluntarily and entirely anonymously, with an informed consent form integrated into the survey's web page. The survey includes 44 questions grouped into four sections as follows:

- Section 1. The first five questions aim to define the profile of the survey participants.
- Section 2. The next group of seven questions helps gather information on the respondents' preferred types of mini-games. Their responses are graded on a Likert scale from one (definitely no) to five (definitely yes).
- Section 3. The third section contains 16 questions to determine the learning style. Here, the popular VARK model [59] was applied that divides learners into four main groups according to their best way of perceiving information. Visual learners prefer to see the information on images, diagrams, or video clips, and the Aural persons learn easier by hearing the educational material in lectures and discussing it. The Read/Write type prefers to read the didactic content from a textbook, take notes, and make lists with the significant points, while Kinesthetic learners remember better by experimenting or movement activities related to the material learned.
- Section 4. The last section also contains 16 questions to determine the playing styles of respondents. These were formulated based on ADOPTA's 40-question survey [60], which determines four playing styles defined on the top of the Kolb theory for experiential learning [61]: Competitor (focused on action, shooting, and competition), Dreamer (enjoys guided gameplay, skill mastering, and reflection), Logician (likes logic, analyses, and contextual thinking), and Strategist (fond of long-term thinking, decision making, and planning strategies). To avoid participants being bored by the lengthy questionnaire, the number of questions was reduced to 16 with responses on the five-level Likert scale. It is essential to clarify that this reduction in the number of questions preserves the reliability of the assessment of the playing style.

Almost all questions' answers in the survey are predefined. The exception is one question in Appendix A.2, where respondents have the opportunity to suggest other puzzle games to be built in the main game. At the end of the questionnaire, participants had the option to check the results for their player and learning styles if they wrote their emails for feedback.

### 4.2. Findings

The data set was constructed from the valid answers collected from 93 students, of which 48 were K12 students and 45 were university students. The mean age of our respondents was circa 15 years, and their gender balance appeared to be 43.01% boys and 56.99% girls. In total, 65.59% of all the respondents reported being excellent students at school.

Together with questions about age, gender, and school grades, the demography section of the survey asked about students' experience in both entertainment and educational

video games. The obtained results demonstrate statistically significant differences between experience in fun gaming and learning games. In total, 39.78% of the participants (*N* = 37) reported playing regularly fun video games (for more than 5 h per week), while there were no regular players of educational games; only 38.71% (*N* = 36) of students shared playing games for learning but less than 5 h weekly. Interesting differences were found in the amount of playing fun games (i.e., commercial games for entertainment) for boys and girls. Figure 5 (left) shows a bar chart of percentages of playing fun video games per week for girls and boys. The T-tests confirm a statistically significant difference (*p* < 0.0000) in the play of commercial games for entertainment reported by girls and boys for all the presented groups and, as well, in general, the play of fun games for girls (*M* = 0.981, *SD* = 1.232, *SE* = 0.171) and boys (*M* = 2.900, *SD* = 1.128, *SE* = 0.176). These differences result in high values of the effect size—Cohen's *d* [62]—which are large than 1.6.

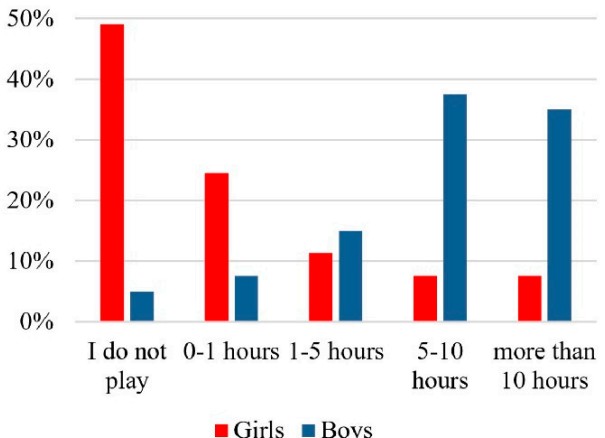 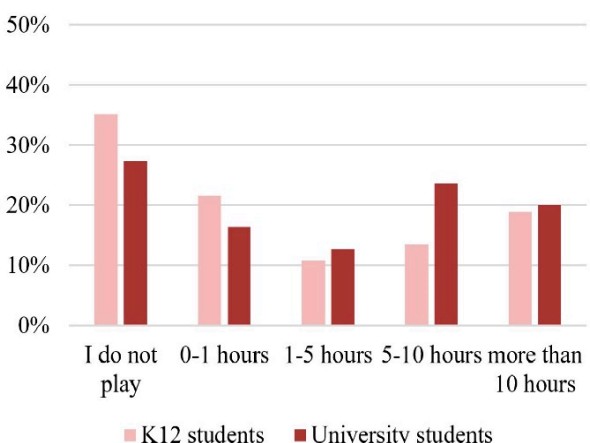

**Figure 5.** Amounts of play of commercial games for entertainment for girls and boys (**left**) and K12 students and university students (**right**).

On the other hand, the differences in playing commercial games for entertainment for K12 and university students are significant only for the group playing such games between 5 and 10 h weekly: 13.51% of K12 students and 23.64% of university students (Figure 5 (right)). At the same time, both girls and boys reported similar ways of learning game playing, as presented in Figure 6 (left), where, for girls, we found *M* = 0.472, *SD* = 0.696, *SE* = 0.097 and for boys *M* = 0.525, *SD* = 0.679, *SE* = 0.106. We found no students playing learning games for more than 5 h per week, as shown in Figure 6. The play amounts of learning games for K12 students and university students appeared similar (see Figure 6 (right)).

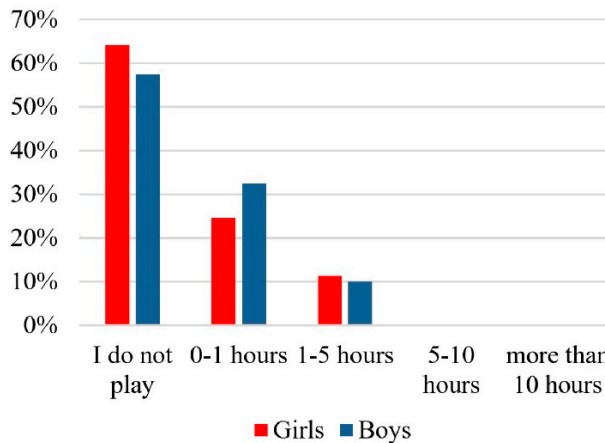 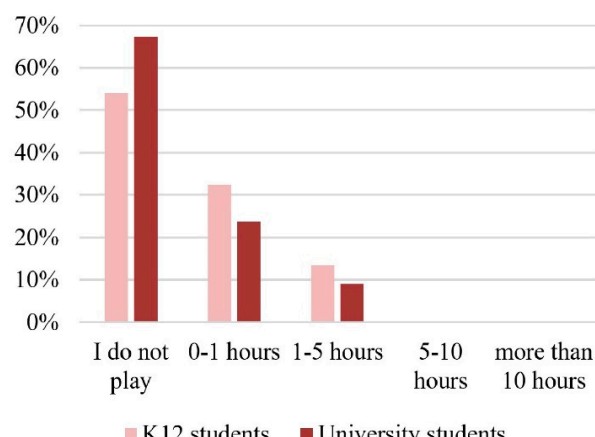

**Figure 6.** Amounts of play of educational games for girls and boys (**left**) and K12 students and university students (**right**).

As expected, the results presented revealed a significant correlation between playing entertainment games and learning games ($r = 0.29106$ at $p < 0.01$). For both girls and boys, paired T-tests show statistically significant differences in playing commercial games for entertainment and educational games ($\Delta M = 1.323$, $p < 0.0000$) with a large effect size (Cohen's $d = 1.122$). However, the difference and the effect size for girls are $\Delta M = 0.509$ ($p < 0.01$) and $d = 0.508$, respectively, while for boys they are $\Delta M = 2.375$ ($p < 0.0000$) and $d = 2.552$, which proves that boys play much more fun games than educational games in respect to girls.

Skewness for the responses to all the questions was between $-1.204$ and $0.425$, while the kurtosis varied from $-0.217$ up to $1.132$. Therefore, the responses have distributions closed to the normal one.

After asking the students about their demography status (Appendix A.1), we asked for their preferences about the types of mini-games included in the educational maze games (Appendix A.2).

Table 1 presents descriptive results about the preferred mini-games for boys and girls and, as well, for K12 students and university students. We found statistically significant differences ($p < 0.05$) with an effect size less than medium (Cohen's $d \approx 0.4$) between gender preferences for both memory games and first-person shooter (FPS) games, proving that boys like these games more than girls do. On the other hand, statistically significant differences were found between the preferences of K12 students and university students for unlocking maze doors ("Open Sesame!") and "Quiz" mini-games ($p < 0.05$, $d \approx 0.5$). Moreover, with greater effect size, statistically significant differences were found for "Divide & Conquer" and "Memory" mini-games ($p < 0.001$, $d \approx 0.8$).

**Table 1.** Results about the preferred mini-games for boys and girls and for K12 students and university students.

| Profile | Boys | | | Girls | | | K12 Students | | | University Students | | |
|---|---|---|---|---|---|---|---|---|---|---|---|---|
| Statistics | M | SD | SE | M | SD | SE | M | SD | SE | M | SD | SE |
| Open Sesame! | 4.00000 | 1.08604 | 0.17172 | 4.22642 | 0.89101 | 0.12239 | 3.83784 | 1.04119 | 0.17117 | 4.29091 | 0.95593 | 0.12890 |
| Quiz | 3.74359 | 1.18584 | 0.18750 | 4.00000 | 1.03775 | 0.14255 | 3.54054 | 1.14491 | 0.18822 | 4.12963 | 1.02876 | 0.13872 |
| 2D puzzle | 3.50000 | 1.17670 | 0.18605 | 3.69811 | 1.24938 | 0.17162 | 3.45946 | 1.14491 | 0.18822 | 3.74545 | 1.25045 | 0.16861 |
| Word soup | 3.61538 | 1.22722 | 0.19404 | 3.67925 | 1.32685 | 0.18226 | 3.59459 | 1.01268 | 0.16648 | 3.68519 | 1.43834 | 0.19395 |
| Roll a ball | 3.23077 | 1.34676 | 0.21294 | 3.16981 | 1.26697 | 0.17403 | 3.05556 | 1.06756 | 0.17551 | 3.24074 | 1.38639 | 0.18694 |
| I see you | 3.41026 | 1.27151 | 0.20104 | 3.39623 | 1.30590 | 0.17938 | 3.22222 | 1.22150 | 0.20081 | 3.50000 | 1.27012 | 0.17126 |
| Find me! | 3.22500 | 1.27073 | 0.20092 | 3.22642 | 1.36778 | 0.18788 | 3.16216 | 1.34399 | 0.22095 | 3.23636 | 1.30474 | 0.17593 |
| Divide & Conquer | 3.84615 | 1.18185 | 0.18687 | 3.84906 | 1.11625 | 0.15333 | 3.32432 | 1.22597 | 0.20155 | 4.18519 | 0.93312 | 0.12582 |
| Memory | 4.23077 | 0.90209 | 0.14263 | 3.84906 | 1.15019 | 0.15799 | 3.45946 | 1.23816 | 0.20355 | 4.31481 | 0.86492 | 0.11663 |
| FPS | 3.75000 | 1.21423 | 0.19199 | 3.22642 | 1.32493 | 0.18199 | 3.18919 | 1.24360 | 0.20445 | 3.61818 | 1.35388 | 0.18256 |

To find the students' preferences regarding the personalization of educational materials, we asked our respondents to rate the importance of choosing didactic content in video games (from four, most important, to one, less important). The mean values ($M$) given in Table 2 show the students prefer individual tailoring of educational materials to be implemented mainly according to initial knowledge in the learning domain, followed by learning goals, age, and finally learning style. The single factor ANOVA (analysis of variance) was applied to test the null hypothesis about equal means. The results show that $F = 10.915043$ while $F_{critical} = 2.63027$; therefore, the null hypothesis was rejected ($p < 0.0000$). T-tests calculated for all the pairs of means proved there are statistically significant differences in mean values for tailoring according to the learning style and according to learning goals and initial knowledge ($p < 0.0000$) and in means for applying age and for initial knowledge and learning style ($p < 0.05$).

**Table 2.** Descriptive statistics for answering the questions about choosing educational materials.

| | Educational Materials in Video Games Should Be Tailored Based on: | | | |
|---|---|---|---|---|
| Statistics | Age | Initial Knowledge | Learning Goals | Learning Style |
| *M* | 2.55682 | 2.88764 | 2.69565 | 2.01149 |
| *SD* | 1.32055 | 0.88470 | 0.99160 | 1.02859 |
| *SE* | 0.13693 | 0.09174 | 0.10282 | 0.10666 |

On the other hand, Figure 7 reveals that preferences for tailoring educational materials depend on the student's age, gaming experience, gender, and grade. Statistically significant differences are found in the mean values for tailoring to learning styles for all pairs of groups except K12 and university students. In addition, there are statistically significant differences in the average values of preferences of players and non-players for tailoring both according to initial knowledge and age. The same is true for the mean values of the preferences of excellent and average students for adaptation to learning objectives.

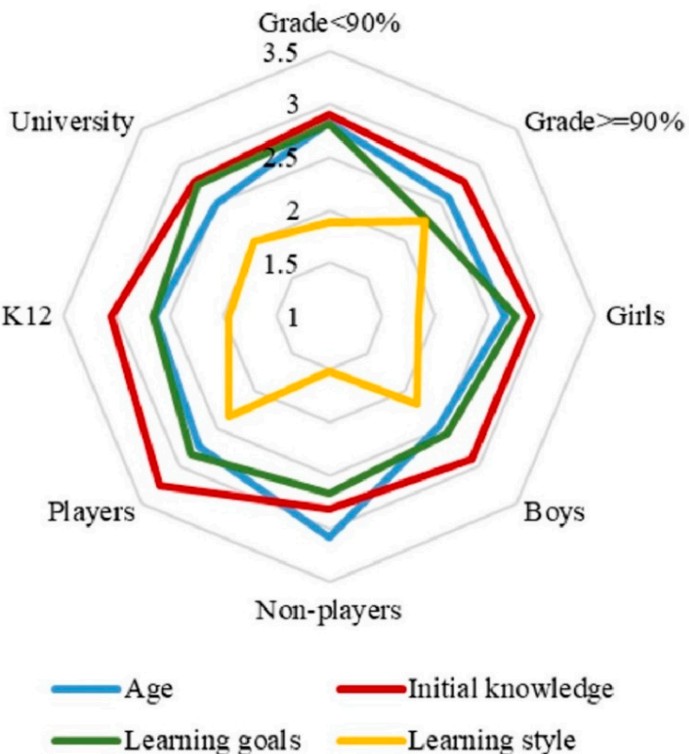

**Figure 7.** Tailoring educational materials according to learning style, learning goals, initial knowledge, and age, separated by age, gaming experience, gender, and grade.

Appendices A.1 and A.2 of our survey did not contain questions asking about the specific construct of the student. In contrast, Appendices A.3 and A.4 had four couples (each containing four items) asking about the VARK learning styles and ADOPTA playing styles, respectively. Therefore, the authors accessed the internal consistency of both the VARK learning style questionnaire and the ADOPTA playing style questionnaire. The found Cronbach's alpha values for the individual components of the VARK learning style were 0.83871 for Visual, 0.84348 for Aural, 0.85765 for Read/Write, and 0.77135 for Kinesthetic, respectively. At the same time, the alpha score for the individual components of ADOPTA playing style was calculated relatively higher: 0.94584 for Competitor, 0.94620 for Dreamer, 0.93690 for Logician, and 0.95664 for Strategist.

The learning and playing styles of the individual students were suggested as a basis for the personalization of educational games [60]. Therefore, we were interested in their influence

on the student-centric tailoring of such video games. As presented in the box plot charts in Figure 8, the predominant learning style from the VARK style family of our respondents (calculated based on self-report from 0.0 up to 1.0) is the Visual one ($M$ = 0.61676, $SD$ = 0.22442, $SE$ = 0.02327). In contrast, the predominant playing style of the ADOPTA style family is Logician ($M$ = 0.75131, $SD$ = 0.12069, $SE$ = 0.01251). The calculated values for skewness and kurtosis assume a distribution closed to the normal one. Outliers lying outside the lower whiskers are available for Dreamer, Logician, and Strategist styles. All the differences in mean values of learning styles and playing styles are statistically significant except for Dreamer ($M$ = 0.67285) and Strategist ($M$ = 0.69607). The statistical significance of the differences is at $p < 0.05$ for Competitor ($M$ = 0.62584) and Dreamer ($M$ = 0.67285) and at $p < 0.0000$ for the other styles.

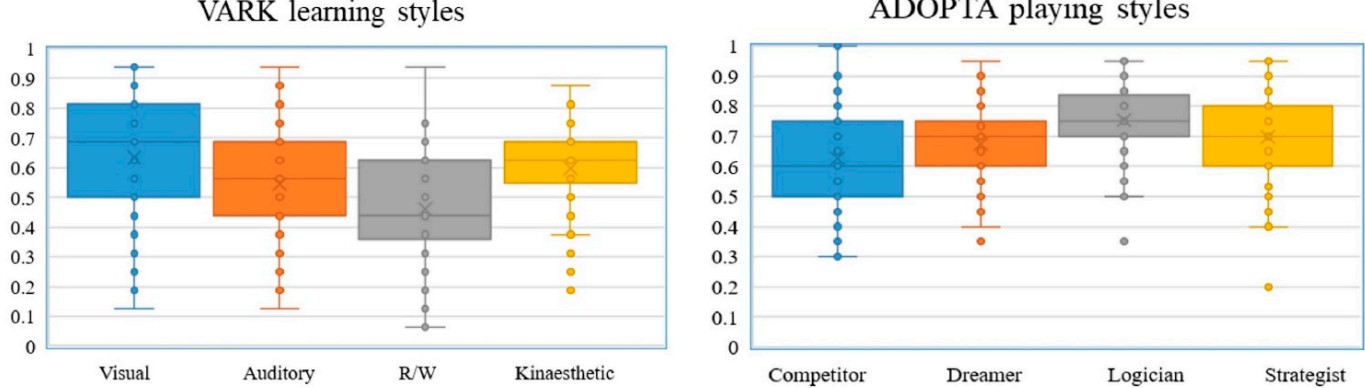

**Figure 8.** Box plot charts of the results found for the VARK learning style (**left**) and ADOPTA playing style (**right**).

The authors conducted a correlational analysis to find eventual dependencies between styles of students and mini-games preferred to be included in maze halls. Findings show that all the learning and playing styles, except Competitor, correlate to the students' preferences for embedding the available mini-games in the educational maze game, as presented in Table 3. Low or medium correlations are identified at different significance levels, marked in bold with one or more asterisks. As a whole, higher correlations are found for VARK learning styles than for ADOPTA playing styles.

**Table 3.** Pearson correlations between results about types of preferred mini-games and individual learning and playing styles.

| Game / Style | Learning Style | | | | Playing Style | | | |
|---|---|---|---|---|---|---|---|---|
| | **Visual** | **Auditory** | **R/W** | **Kinesthetic** | **Competitor** | **Dreamer** | **Logician** | **Strategist** |
| Open Sesame! | 0.12508 | 0.20003 | 0.16335 | 0.19464 | −0.09837 | 0.13555 | 0.17293 | 0.14964 |
| Quiz | 0.13541 | **0.31965** ** | **0.24366** * | **0.21416** * | −0.02687 | 0.06410 | 0.00110 | −0.09324 |
| 2D puzzle | **0.45340** **** | **0.37464** *** | **0.25499** * | **0.22134** * | 0.10702 | **0.35769** *** | **0.32623** ** | **0.29900** ** |
| Word soup | **0.28891** ** | **0.26010** * | 0.17173 | **0.26171** * | −0.03246 | **0.20510** * | **0.33249** ** | **0.25796** * |
| Roll-a-ball | **0.49104** **** | **0.21975** * | 0.18092 | **0.30027** ** | −0.05383 | **0.23836** * | **0.30194** ** | **0.28186** ** |
| I see you | **0.27400** ** | 0.15509 | **0.33271** ** | **0.36235** *** | 0.04655 | **0.21694** * | **0.32653** ** | 0.18531 |
| Find me | 0.15645 | 0.16660 | 0.17388 | **0.26215** * | 0.17911 | 0.05385 | 0.10386 | 0.03895 |
| Divide & Conquer | 0.17675 | 0.07566 | 0.11695 | **0.35028** *** | 0.05816 | 0.06690 | **0.24277** * | 0.08856 |
| Memory | **0.27721** ** | 0.06770 | 0.08237 | **0.25673** * | 0.04362 | 0.03626 | 0.16390 | **0.20191** * |
| FPS | **0.26740** ** | **0.26740** ** | **0.22373** * | **0.24859** * | 0.08085 | 0.16789 | 0.11203 | **0.24433** * |

\* $p < 0.05$. \*\* $p < 0.01$. \*\*\* $p < 0.001$. \*\*\*\* $p < 0.0001$.

Any statistically significant correlations were not found between the answers about tailoring educational materials and learning or playing styles. However, significant correlations were identified between the player's propensity to replay a mini-game (to improve the score for that level of difficulty) and Dreamer, Logician, and Strategist playing styles, as presented in Table 4. Moreover, both Logician and Strategist correlate with the inclination to play the mini-game at the next level of difficulty.

**Table 4.** Pearson correlations between results about replaying at the same level and playing at a higher level, the same mini-game, and individual learning and playing styles.

| Game Option \ Style | Learning Style | | | | Playing Style | | | |
|---|---|---|---|---|---|---|---|---|
| | Visual | Auditory | R/W | Kinesthetic | Competitor | Dreamer | Logician | Strategist |
| Replay at the same level | 0.12995 | −0.06889 | 0.10200 | 0.08292 | 0.07951 | **0.24940 ** | **0.33031 *** | **0.31136 *** |
| Play at a higher level | 0.07820 | −0.02088 | 0.09932 | 0.07082 | 0.01028 | 0.19579 | **0.39762 **** | **0.39654 **** |

** $p < 0.01$. *** $p < 0.001$. **** $p < 0.0001$.

Finally, Table 5 presents the Pearson correlations between all the playing and learning styles. As expected, VARK learning styles and ADOPTA playing styles correlate highly within the style family. On the other hand, we found statistically significant correlations between the Visual learning style and Dreamer and Strategist. The same is for the R/W learning style and Logician and Strategist playing styles.

**Table 5.** Pearson correlations between learning and playing styles.

| Style \ Style | Learning Style | | | | Playing Style | | | |
|---|---|---|---|---|---|---|---|---|
| | Visual | Auditory | R/W | Kinesthetic | Competitor | Dreamer | Logician | Strategist |
| Visual | **1.00000** | **0.48089 **** | **0.47543 **** | **0.37190 *** | 0.02744 | **0.24279 * | 0.19545 | **0.27290 * |
| Auditory | | **1.00000** | **0.34884 *** | **0.45241 **** | 0.00437 | 0.10707 | 0.02441 | 0.14558 |
| R/W | | | **1.00000** | **0.26148 * | −0.13917 | 0.20318 | **0.21070 * | **0.22035 * |
| Kinesthetic | | | | **1.00000** | 0.13670 | −0.05453 | 0.07392 | 0.08745 |
| Competitor | | | | | **1.00000** | −0.03609 | 0.01874 | 0.04855 |
| Dreamer | | | | | | **1.00000** | **0.34665 *** | **0.27263 * |
| Logician | | | | | | | **1.00000** | **0.55435 **** |
| Strategist | | | | | | | | **1.00000** |

* $p < 0.05$. *** $p < 0.001$. **** $p < 0.0001$.

## 5. Discussion

Today's students are from the digital generation that needs alternative teaching strategies, including engaging and active learning, to be provided to them to make learning more attractive and motivating. Among other approaches, these strategies include serious games, and especially puzzle-based learning, which have recently been reinvented in teaching practice. They are proved to be more learner-centered, involving, interactive, and efficient than traditional teaching. In addition, they are appropriate for many subject areas [63]. Much research has discussed the positive effects on students of including different types of puzzles in lessons [64,65].

There are many studies in the area of educational puzzle games, but none of them has explored the student preferences about learning through such games. In addition, there are no current studies on the application of such puzzles as mini-games in a maze, which is a completed and unified educational video game of a maze type. We argue that such a study will contribute to improving the design and content of these games. In previous studies by the authors of this article, the user experience and the satisfaction of the learner from playing the educational video maze game "Let Us Save Venice" [12], generated on the APOGEE software platform, were thoroughly studied. The results generally proved the validation of the game, the advantages of the mini-games, and the educational and game content included in the maze, as well as the high overall assessment of the user experience and the game by the users. However, we did not find research on how to choose an appropriate type of puzzles (here called mini-games) according to the profiles of students. We have explored this issue in the scope of the personalization framework and presented the survey findings in the current article.

### 5.1. Discourse on Findings Regarding the Research Questions

In this study, we tried to find the characteristics of the student's model that should be considered for the personalization of educational video games. As presented by the results

in the previous section, it was concluded that boys, compared to girls, play much more commercial games for entertainment than educational games (Figure 5). About 50% of the respondents do not play fun games, while the result is only about 5% for boys who do not play entertainment games. The results are similar in educational games, as about 60% of respondents do not play educational games. Summarizing these results, it turns out that these results are not accidental, precisely because fun games are much more accessible and widespread than educational games. Some of the results regarding the preferences of the respondents for the different mini-games revealed a significant difference in the attitude between boys and girls.

As a whole, there are high percentages for boys and girls regarding their play on both fun and educational games at different degrees of frequency. These positive results show the upward trend in the use of educational video games. In this regard, of all respondents ($N$ = 93), it appears that the K12 students play more educational games than commercial games for entertainment, while the opposite is true for university students. University students play more entertainment games than educational games. This fact reaffirms that educational games in the learning process are helpful for students, although these games are available to consumers to varying degrees. More university students will play such educational games if more effort and resources are put into developing educational games at universities.

Next, the authors tried to find the student preferences regarding the personalization of educational video games. Some mini-games such as "Roll a Ball", "Find me!", "I see you", and "FPS" (with $M < 3.5$) are less preferred than others such as "Open Sesame!", "Memory", "Divide & Conquer", and "Quiz" (with $M > 3.8$). This trend is much more visible among the university students than K12 students ($\Delta M = 0.45$). While the K12 students prefer mostly "Open Sesame!", "Word soup", and "Quiz" mini-games (all with $M$ greater than 3.5), university students prefer mostly "Memory", "Open Sesame!", "Divide & Conquer", and "Quiz" (all with $M$ greater than 4). According to the results, all students prefer the question games type. The younger of them (K12) also like to play the "Word soup" game that needs observation. On the other hand, the senior students chose games for developing memory and associating objects with a common feature. Surprisingly, the students do not appreciate the action game type and the games for searching.

The K12 students and game players mostly prefer tailoring educational materials according to initial knowledge. That could be explained by their desire to apply the knowledge they already have and acquire new ones. Growing mature, the university students realize the significance of different learning goals, and for this reason, they prefer adapting the learning content according to them. On the other hand, the non-players consider age to be the most relevant base for tailoring educational materials because they do not have experience with video games. For this reason, they do not easily find the relation between playing and increasing knowledge, achieving learning goals, or applying learning styles. The results also show that the K12 and university students do not appreciate the possibility of tailoring educational materials according to learning styles. The reason for this is that they are still not familiar enough with these styles and cannot adequately assess the differences between them.

The school marks also are a factor for students' preferences concerning the personalization of mini-games. Excellent students with grades over 90% find learning goals less significant for tailoring the built-in didactic content, while initial knowledge is essential for content personalization (Figure 7). Such findings confirm the common belief that excellent students usually are more focused, so they prefer to play games offering the possibility to acquire knowledge. For students with average marks below 90%, initial knowledge, learning goals, and age are almost equally substantial for tailoring educational materials. The vast difference in preferences concerns the learning style. Non-excellent students do not think it matters, while the others believe that it and learning goals have to be taken into account during the customization process of mini-games with an average significance level.

Figure 7 shows that learning style as a reason for personalization has a different level of importance for all groups of students but, as a whole, is the less preferable feature according to which the educational material in the maze game is to be tailored. The large discrepancy is between players and non-players and between excellent and average students. The possible reason is that, usually, students are not aware enough of their learning styles and cannot adequately assess how learning styles can be addressed in the learning process.

Despite some variations in the level of preferences, the results in Figure 7 show that all the students (excluding non-players) prefer the personalization of mini-games to be made considering their initial knowledge, learning goal, age, and learning style in that sequence. It is essential to outline those students who do not have playing experience and consider age as the most relevant parameter for learning content customization. In contrast, they almost neglect learning style for such a purpose.

Correlations found between preferred types of mini-games and individual learning and playing styles (Table 3) show that the personalization of learning content and placement of 2D and 3D mini-games in the halls of the maze can be carried out based on the style of learning and playing. Of all the styles, only Competitor did not show a statistically significant correlation with mini-games, which necessitates further research on the preferences of this style. For the practical implementation of style personalization, it is essential to decide how the dominant style (or styles) of learning and playing will be determined and, next, how the style personalization will be combined with other types of personalization.

The third research question was about the organization of the personalization process of educational video games. The results displayed in Table 2 and Figure 7 suggest that the tailoring of educational materials (such as didactic content on the learning boards and educational mini-games in maze halls) should be performed first based on initial knowledge in the learning domain (e.g., beginner, intermediate, and expert), followed by learning goals, age, and finally by learning style. At each of these personalization steps, we should bear in mind the mini-games preferences for boys and girls and K12 students and university students (shown in Table 1), together with the statistically significant biases on age, gaming experience, gender, and grade (presented in Figure 7). In general, didactic content in maze games could be tailored based on age, learning goals, knowledge level, and learning style. At the same time, gameplay (i.e., the choice of mini-games types) could be personalized based on age, gender, and learning and playing style. This personalization is based on static features of the student model. It is supposed to be carried out within the individual game instance before starting the game session. At the same time, dynamic adjustment of task difficulty, mini-game feedback and hints, the behavior of non-player characters, and audio-visual details should be accomplished at run time during the game, based on playing outcomes and the emotional state of the student.

The internal consistency of the VARK learning style questionnaire was found to be similar to the score reported in [66]. On the other hand, Cronbach's alpha score found for the individual components of the ADOPTA playing style questionnaire appeared to be relatively higher than the one reported in [60] for ADOPTA's survey containing 40 items with dichotomous response options. The higher reliability of the current survey is because the previous research [60] applied a sequence of 10 items with dichotomous response options for each of the concepts. The dichotomous responses resulted in alpha values of 0.63606 for Competitor, 0.80414 for Dreamer, 0.78935 for Logician, and 0.70201 for Strategist. In this study, the authors reduced each sequence of 10 items per concept to four items with response options applying a five-level Likert scale, which resulted in higher reliability of the new questionnaire.

### 5.2. Limitations of the Study

The present article has the following limitations in summarizing the presented study and the practical experiments conducted. A limitation of the study is that the investigation has a relatively small number of respondents ($N$ = 93). Of course, this study can be applied repeatedly and thus gather many more respondents. Another limitation is that the

respondents have an uneven distribution between K12 and university students. This fact, to some extent, limits the research, and a balance is not reached between the two groups of students. Another limitation of the study is that the game session has not yet been registered in the system. So, the students watched a video and were instructed to play the educational game. However, they were not required to play the game before completing the questionnaires. The platform has not yet been finalized, so we were unable to register the exact number of mini-games played, points earned, playing time, and results.

Another limitation of the study is that these results are obtained as self-reported data (questionnaires filled voluntarily and only by users) without significant evidence that the information is 100% correct and that the respondents completed the questionnaires conscientiously and responsibly. In addition, a study limitation is that 65.59% of all the respondents report having excellent grades, but this is not representative of students with medium grades.

## 6. Conclusions and Future Works

Educational maze video games remain very appealing and engaging, as they can embed various types of learning resources. Usually, mazes have a regular structure and therefore are suitable for a formal description and an automatic generation based on it. The authors apply 3D video mazes as containers of both learning resources and various 2D and 3D puzzle mini-games. Further, they enable a different design in each hall, which can be related to the context of mini-games and contribute to the students' immersion and a more engaging learning experience. They are especially appropriate for presenting, exploring, and testing knowledge because they can integrate different media (text, images, audio, and video) that represent learning content. The present article described the student preferences in tailoring such educational maze video games. Video game personalization in a learning context based on the student model was briefly explained. The formulated research questions considered both students' model characteristics and student preferences as a base for personalizing educational video games. Therefore, the process of personalization of educational video games should use the profile of the students for making choices or learning resources and mini-games embedded in maze halls.

The focus of the study is on the idea of student-centered tailoring of educational games. The maze games generated by the APOGEE project platform apply the outlined personalization framework with three groups of parameters:

- Demographics, preferences, and goals.
- Learning and playing styles.
- In-game measured performance and efficiency.

The authors also described a student's model based on static and dynamic features of the user's, learner's, and player's aspects. Further, the personalization process of educational games using the student's model was summarized. The article also presented the platform-generated maze game with the possibilities of personalization of its mini-games divided into four groups according to the actions required by players for successful passing.

A survey to determine the participants' learning and play styles according to the VARK and ADOPTA models was conducted for the research purposes. The results of this survey show statistically significant differences between playing video games for fun and learning. It turns out that about 40% of respondents play entertainment games more than 5 h a week, but no one plays educational games for so much time. Boys play video games for fun much more often than girls do. On the other hand, the results for K12 and university students are very similar.

The students also indicated what mini-games they prefer to play within the main maze game. There are statistically significant differences between the "Memory" and "FPS" games from the points of view of boys and girls. In general, university students appreciate the proposed mini-games more than K12 ones, the most significant difference being for "Divide & Conquer", "Memory", "Open Sesame!", and "Quiz" mini-games.

According to the respondents, the educational material should follow the initial knowledge in the learning domain and the players' ages and at least with their learning styles. The correlations for VARK styles are higher compared to ADOPTA styles. It turns out that the Visual learning style correlates to Dreamer and Strategist, and the Read/Write learning style correlates to Logician and Strategist playing styles. The reasons for obtaining results are discussed in detail, and the future works are outlined.

Our future works will include the generation of both personalized and non-personalized versions of the same educational game to conduct practical experiments with them. By measuring the game outcomes, learnability, and playability for both the experimental and control student groups (playing these versions of the same game), we will validate the expected benefits of student-centric personalization of both the educational content and gameplay. Similar experiments will be conducted with both adaptive and non-adaptive versions of the same game aiming to validate dynamic adjustment of task difficulty and NPC behavior based on changes in game outcomes and the emotional state of individual students.

**Author Contributions:** Conceptualization, V.T., B.B., Y.D. and E.P.-H.; methodology, V.T., B.B., Y.D. and E.P.-H.; validation, V.T., B.B., Y.D. and E.P.-H.; writing—original draft preparation, V.T., B.B., Y.D. and E.P.-H.; writing—review and editing, V.T., B.B., Y.D. and E.P.-H.; visualization, V.T., B.B., Y.D. and E.P.-H.; supervision, B.B.; project administration, B.B.; funding acquisition, B.B. All authors have read and agreed to the published version of the manuscript.

**Funding:** The research leading to these results received funding from the APOGEE project, funded by the Bulgarian National Science Fund, grant agreement No. DN12/7/2017.

**Institutional Review Board Statement:** The study was conducted according to the guidelines of the Ethical Code of Sofia University and follows the EU recommendations and approved standards in the field.

**Informed Consent Statement:** Informed consent was obtained from all subjects involved in the study.

**Data Availability Statement:** Not applicable.

**Conflicts of Interest:** The authors declare no conflict of interest.

## Appendix A. Questionnaire

*Appendix A.1. Section 1*

1.  What is your age (in years)?
2.  What is your sex?

    a.  Male
    b.  Female

3.  What is (or has been) your average success at school?

    a.  Fair
    b.  Good
    c.  Very good
    d.  Excellent

4.  How many hours a week do you play computer video games?

    a.  I do not play
    b.  0–1 h
    c.  1–5 h
    d.  5–10 h
    e.  More than 10 h per week

5.  How often do you play computer games related to the learning material?

    a.  I do not play
    b.  0–1 h per week
    c.  1–5 h per week
    d.  5–10 h per week

e. More than 10 h per week

*Appendix A.2. Section 2*

1. The educational video games you would like to play, what should they be directed at?

    a. Initially acquaintance with the teaching theme
    b. Performing experiments on the teaching topic
    c. A detailed study of the teaching topic
    d. Testing knowledge on the teaching topic
    e. The revision and summarization of the teaching topic
    f. Links between the teaching theme of the game and other study topics

2. It is a good idea to include the following types of mini-games in the educational video maze games:

    a. Answer a question to unlock the door to another hall in the maze ("Open Sesame!")
    b. Answer a few questions from the Game Learning Area ("Quiz")
    c. Solving a 2D puzzle with an educational image ("2D puzzle")
    d. Solving a puzzle with words—for example, to find words from the study area of the game in a table of letters ("Word soup")
    e. Rolling balls marked with text/picture to the correct objects or positions on the floor map ("Roll a ball")
    f. Discovering and collecting visible translucent objects/objects ("I see you")
    g. Discovering and collecting invisible objects/objects hidden in larger visible objects by moving large objects ("Find me!").
    h. Grouping of objects/objects by a given sign ("Divide & Conquer")
    i. Memory Development Game—In a matrix of hidden words/pictures to find those who are two-by-two identical ("Memory").
    j. Shooting on moving inanimate objects, such as balloons with an attached study object (First Person Shooter, or FPS)

3. What other learning mini-games and/or tasks would you offer to include in the educational maze game? (Tell us your opinion in free text)

4. Rank in importance how to select learning materials in educational video games:

    a. Selected according to the player's age
    b. Selected to the player's level of knowledge in the study area of the game (eg beginner, advanced and expert)
    c. Selected according to the interests and goals of the player (Initial familiarization with the topic, detailed study, testing of knowledge, and so on)
    d. Selected according to the predominant learning style of the learner (Visual, Aural, Read/Write, and Kinaesthetic)

5. Rank in importance how to automatically adjust the difficulty of playing educational video games:

    a. According to the emotions/feelings and excitement of the player
    b. According to the player's playing result
    c. According to the prevailing playing style of the player (Competitor, Dreamer, Logician, and Strategist)

6. Would you play an educational game once again at the same level of difficulty to improve the result for this level?

7. Would you play an educational game at the next level of difficulty?

*Appendix A.3. Section 3*

(VARK Questionnaire [54]).

*Appendix A.4. Section 4*

Applying the 5-point Likert scale—Strongly disagree (1), Disagree (2), Neutral (3), Agree (4), Strongly Agree (5).

1. When I play, I often take great risks and rely mainly on my intuition (in my inner voice) instead of thinking and analysis.
2. I want to be able to play at a certain level in the game until I master it enough.
3. I like the logical approaches to analysing the actions in the game to come up with successful tactics and playing strategies.
4. I want to solve practically complex problems in the game on time, easily and in the most effective way.
5. I prefer to solve problems spontaneously, relying on my composure in critical situations and "sample and error" methods, without much considering or discussing the consequences of one or the other solution.
6. I don't like time limitations and I want to observe and think as long as I need.
7. I study the complexity of every rule, as well as the facts and actions in the game, to use them in a reasonable and most useful way.
8. I prefer to start actions in the game only with reasonable expectations for practical results and benefits.
9. I prefer to start actively playing as soon as possible without reading instructions or planning in advance.
10. I prefer to watch, listen to and consider the arguments of others to clarify the script of the game before making decisions and starting playing actively.
11. When discussing with other players, I do not trust the arguments and assumptions of others, but prefer to check and test everything myself.
12. I think long-term and plan such game strategies that can quickly achieve practical results.
13. In the team game, I am considered the most active player, and in discussions, I prefer to talk and share my achievements with other players.
14. When I play, I show what I feel, and in the discussions, I participate less than the others.
15. I like to be recognized by others as a thinking, consistent and fair player.
16. I have good organizational skills and I want to command and lead the team in the game process.

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
