# Peer review of "How to Tailor Educational Maze Games: The Student’s Preferences"

_sustainability, doi:10.3390/su14116794_

Round 1

Reviewer 1 Report

This paper is well-written (minor English problems), and presented in an academic standard. However, I have raised a few comments I believe would strengthen the manuscript before publication:

  1. Further explanation is required about ‘static’ and ‘dynamic’ features (lines 286-294). Some statements are unclear or sound repetitive, for example ‘initial knowledge in the learning domain, learning, and playing goals’ (line 288), the second ‘learning’ here refers to the learning progress? Learning outcome?

  1. Some statements need to be more specific, for example ‘personalization is based on the view that variety (sic) in some characteristics of users can impact the usability and efficiency of the services provided to them’ (lines 307-308). What do the authors mean by ‘variety’ in this sentence?

  1. Figure 3. Why two factors (age and learning style) are repeated in the selection of ‘learning content’ and the ‘selection of mini-games’ while the others are not?

  1. Survey results. The authors should better explain the method and instruments before the survey results. For example, 4.1 Methodology should be previously explained, including context, sampling type, research stages (survey administration) and instrument/s.

  1. The main problem I see in this paper is related with the instrument (survey), which is said to be based on a survey comprising 44 questions. This is not a validated scale available in previous works, so it should be included in an Appendix. The authors refer to some limitations (lines 692-694) related with the format as this is mostly a self-reported scale. So, what measures were taken to avoid some statistical bias related with these type of scales (acquiescence bias, reverse coding, etc)? As the survey was not included in the paper determining its reliability can be difficult.

  1. Findings. Why do the authors distinguish between ‘fun games’ and ‘learning games’ (line 476)? Do they mean ‘commercial games’ and ‘educational or serious games’. What is this classification fun/learning games based on? They authors should also provide examples of ‘fun’ versus ‘learning’ games.

  1. I suggest changing the format of figures 6 and 7 (bar charts) into tables as in Table 1 for professional reasons. Figures 6 and 7 as tables should also include Standard Deviation (SD).

  1. Some statements need further clarification. For example, ‘it is interesting to note that K12 students play more educational games than fun games, while the opposite is true for university students’ (line 600). This is the discussion section, so why do the authors think this is true rather than saying ‘it is interesting’? And the authors should avoid repeating the same statement, for example ‘it is interesting to note that K12 students play more educational games than fun games, while the opposite is true for university students. University students play more fun games than learning games. (lines 600-601)

  1. Discussion and Conclusions. No reference is made to the results from previous works. The authors should discuss their results by comparing them to other related works and highlight the novelty of their findings.

Author Response

Response to Reviewer 1

Dear Reviewer,

Thank you very much for your time and effort spent reviewing our submission with Manuscript ID equal to "sustainability-1728451". We have thoroughly reflected in the new version of the paper all of your remarks and proposals. Here go our answers to each one of them (numbers of lines are from the revised version of the article):

1. Further explanation is required about ‘static’ and ‘dynamic’ features (lines 286-294). Some statements are unclear or sound repetitive, for example ‘initial knowledge in the learning domain, learning, and playing goals’ (line 288), the second ‘learning’ here refers to the learning progress? Learning outcome?

Revision made: We have made the necessary explanations about static and dynamic features in the model of students. The text should look clearer and more detailed in this section of the newer version of the article (lines 312-320, 349-363).

2. Some statements need to be more specific, for example ‘personalization is based on the view that variety (sic) in some characteristics of users can impact the usability and efficiency of the services provided to them’ (lines 307-308). What do the authors mean by ‘variety’ in this sentence?

Revision made: We have made the necessary changes in Section 3.2, including adding more text in this paragraph as follows below. The text should look clearer and more detailed, at this section of the newer version of the article (lines 339-347).

3. Figure 3. Why two factors (age and learning style) are repeated in the selection of ‘learning content’ and the ‘selection of mini-games’ while the others are not?

Revision made: We explained in detail why the selection of ‘learning content’ and the ‘selection of mini-games’ use both age and learning style and, as well, we commented on why we apply the other features (363-371).

4. Survey results. The authors should better explain the method and instruments before the survey results. For example, 4.1 Methodology should be previously explained, including context, sampling type, research stages (survey administration) and instrument/s.

Revision made: The methodology was extended with more details by adding information, including context (European Researchers' Night in Bulgaria in 2020 and 2021), sampling type (K12 and university students), research stages (online survey administration), and instrument/s (voluntarily and anonymous participation, with an informed consent form) – lines 521-541. As well, we briefly explained both the VARK learning style family and the ADOPTA playing style family (lines 548-565).

5. The main problem I see in this paper is related with the instrument (survey), which is said to be based on a survey comprising 44 questions. This is not a validated scale available in previous works, so it should be included in an Appendix. The authors refer to some limitations (lines 692-694) related with the format as this is mostly a self-reported scale. So, what measures were taken to avoid some statistical bias related with these type of scales (acquiescence bias, reverse coding, etc)? As the survey was not included in the paper determining its reliability can be difficult.

Revision made: The questionnaire (consisting of four sections and comprising 44 questions) is included now in an Appendix (lines 949-1053). Its sections A and B did not contain questions asking about the specific construct of the student, while sections C and D had four couples (each one containing four items) asking about each VARK learning style and ADOPTA playing style, respectively. In order to assess the internal consistency and reliability of the questionnaires C and D, we evaluated Cronbach's alpha values for each couple of four items (lines 658-667).

6. Findings. Why do the authors distinguish between ‘fun games’ and ‘learning games’ (line 476)? Do they mean ‘commercial games’ and ‘educational or serious games. What is this classification fun/learning games based on? The authors should also provide examples of ‘fun’ versus ‘learning’ games.

Revision made: We explained explicitly that fun games are commercial games for entertainment, and as well, that learning games are educational games. As well, we replaced their occurrences – “fun games” was replaced by “commercial games for entertainment” and “learning games” was replaced by “educational games”.

7. I suggest changing the format of figures 6 and 7 (bar charts) into tables as in Table 1 for professional reasons. Figures 6 and 7 as tables should also include Standard Deviation (SD).

Revision made: The authors suggest keeping the format of fig.6 because it is strongly related to fig. 5 and, therefore, should be preserved as a figure. The authors accepted the proposal to present the data in fig. 7 as a new table, together with SD and SE (lines 661-632). All the following numbers of figures and tables were re-enumerated in order to cite Tables and Figures in correct order. As well, we placed the Tables and Figures closely behind their first citations in the main text.

8. Some statements need further clarification. For example, ‘it is interesting to note that K12 students play more educational games than fun games, while the opposite is true for university students’ (line 600). This is the discussion section, so why do the authors think this is true rather than saying ‘it is interesting’? And the authors should avoid repeating the same statement, for example ‘it is interesting to note that K12 students play more educational games than fun games, while the opposite is true for university students. University students play more fun games than learning games. (lines 600-601)

Revision made: The phraseology in the discussion section has been improved for clarification (lines 757, 890)

9. Discussion and Conclusions. No reference is made to the results from previous works. The authors should discuss their results by comparing them to other related works and highlight the novelty of their findings.

Revision made: We referred to our previous research. We have added text that indicates that this study focuses on an unexplored area, namely the preferences of consumers. In our previous research, the user experience of an analogous game generated on the APOGEE software platform was thoroughly examined, with the addition of some new references (lines 720-741).

The authors accessed the internal consistency of both the VARK learning style questionnaire and the ADOPTA playing style questionnaire. Cronbach’s alpha scores for both the VARK learning style questionnaire and the ADOPTA playing style questionnaire were compared to scores found in previous studies (lines 838-848).

We remain at your disposal for any further comments and suggestions!

The authors

24 May 2022

Reviewer 2 Report

1. The research topic and key issues discussed in the present paper are emergent.

2. Literature review is logical and focused.

3. Research methods and process are clear and well-organised.

4. Findings and results are able to respond to the key research questions.

5. In the discussion part, the authors are required to compare their findings with the related studies and to show whether the findings support previous studies based on literature review.  

Author Response

Response to Reviewer 2

Dear Reviewer,

Thank you very much for your time and effort spent reviewing our submission with Manuscript ID equal to "sustainability-1728451". We have thoroughly reflected in the new version of the paper all of your remarks and proposals. Here go our answers to each one of them (numbers of lines are from the revised version of the article):

  1. The research topic and key issues discussed in the present paper are emergent.

Revision made: None.

  1. Literature review is logical and focused.

Revision made: Thank you for your appreciation of our literature review. We added additional text about applications of serious games, with some new references (lines 49-72). We discussed how serious games can address the main learning goals concerning sustainability issues, together with the role of serious games in sustainability education - e.g., for increasing learners’ awareness of the school subjects and crucial issues like healthcare, preserving the culture and historical heritage, climate changes, social causes, etc.

  1. Research methods and process are clear and well-organised.

Revision made: None.

  1. Findings and results are able to respond to the key research questions.

Revision made: None.

  1. In the discussion part, the authors are required to compare their findings with the related studies and to show whether the findings support previous studies based on the literature review.  

Revision made: We referred to our previous research. We have added text that indicates that this study focuses on an unexplored area, namely the preferences of consumers. In our previous research, the user experience of an analogous game generated on the APOGEE software platform was thoroughly examined, with the addition of some new references (lines 720-741).

The authors accessed the internal consistency of both the VARK learning style questionnaire and the ADOPTA playing style questionnaire. Cronbach’s alpha scores for both the VARK learning style questionnaire and the ADOPTA playing style questionnaire were compared to scores found in previous studies (lines 838-848).

We remain at your disposal for any further comments and suggestions!

The authors

24 May 2022

Reviewer 3 Report

It is an interesting work and a theme that I like very much in view of the need to change the teaching paradigm and invest in an action learning approach. However, despite the theme being relevant, several questions and proposals for improvement arise:

- As I highlighted earlier, I consider the topic to be scientifically relevant in the field of education. However, the alignment with the scope of the Sustainability journal is somewhat doubtful. In fact, throughout the paper there is no reference to the theme of sustainability. Connecting the area of study with the factors related to sustainability in education is essential. In particular, I suggest that the authors address the role of serious games in sustainability.

- Authors note “The data set was constructed from the valid answers collected from 93 students, whereupon 48 of them were K12 students and 45 were university students. The sample is relatively small. However, my main concern is about the heterogeneous profile of these students. Are the games played by university students and k12 students exactly the same? Are they suitable for both ages?

- How is each student's initial knowledge measured? Through a questionnaire? Or is there a hybrid approach?

- I recommend the authors to present the full structure of the survey in annex.

- Authors should better describe and contextualize the several different learning styles.

- The discussion of results is weak. Only a discussion based on the previously defined research questions is made. This is an important, but insufficient, point. It would be desirable if the discussion of results also compared the results obtained by other studies in the literature.

- There are several approaches to building serious games. Maze games is just one of them. It would be important to understand when this approach is a better fit compared to other typologies of serious games. This is a topic little explored by the authors. Is it the profile of the students that should justify this choice or type of learning outcomes?

Author Response

Response to Reviewer 3

Dear Reviewer,

Thank you very much for your time and effort spent reviewing our submission with Manuscript ID equal to "sustainability-1728451". We have thoroughly reflected in the new version of the paper all of your remarks and proposals. Here go our answers to each one of them (numbers of lines are from the revised version of the article):

- As I highlighted earlier, I consider the topic to be scientifically relevant in the field of education. However, the alignment with the scope of the Sustainability journal is somewhat doubtful. In fact, throughout the paper there is no reference to the theme of sustainability. Connecting the area of study with the factors related to sustainability in education is essential. In particular, I suggest that the authors address the role of serious games in sustainability.

Revision made: We added additional text about applications of serious games, with some new references (lines 49-72). We discussed how serious games can address the main learning goals concerning sustainability issues, together with the role of serious games in sustainability education - e.g., for increasing learners’ awareness of the school subjects and crucial issues like healthcare, preserving the culture and historical heritage, climate changes, social causes, etc.

- Authors note “The data set was constructed from the valid answers collected from 93 students, whereupon 48 of them were K12 students and 45 were university students. The sample is relatively small. However, my main concern is about the heterogeneous profile of these students. Are the games played by university students and k12 students exactly the same? Are they suitable for both ages?

Revision made: In order to differentiate student playing preferences according to age, both university and K12 students played exactly the same maze game and, next, responded to the same questions about their preferences of mini-games included in the maze (as additionally explained in 4.1 – lines 535-541).

- How is each student's initial knowledge measured? Through a questionnaire? Or is there a hybrid approach?

Revision made: Initial knowledge is self-assessed before the game session whereupon the student has to choose one of several stereotypes such as beginner, advanced, and expert. For learning goals, the student can opt between an introduction to the subject, a game with experiments, a detailed study, an assessment game, and a summarization (see Section 3.2, lines 349-371).

- I recommend the authors to present the full structure of the survey in annex.

Revision made: The questionnaire (consisting of four sections and comprising 44 questions) is included now in an Appendix (lines 949-1053).

- Authors should better describe and contextualize the several different learning styles.

Revision made: in section 4.1, the authors have outlined both the VARK learning styles and the ADOPTA playing styles (lines 548-565).

- The discussion of results is weak. Only a discussion based on the previously defined research questions is made. This is an important, but insufficient, point. It would be desirable if the discussion of results also compared the results obtained by other studies in the literature.

Revision made: We referred to our previous research. We have added text that indicates that this study focuses on an unexplored area, namely the preferences of consumers. In our previous research, the user experience of an analogous game generated on the APOGEE software platform was thoroughly examined, with the addition of some new references (lines 720-741).

The authors accessed the internal consistency of both the VARK learning style questionnaire and the ADOPTA playing style questionnaire. Cronbach’s alpha scores for both the VARK learning style questionnaire and the ADOPTA playing style questionnaire were compared to scores found in previous studies (lines 838-848).

- There are several approaches to building serious games. Maze games is just one of them. It would be important to understand when this approach is a better fit compared to other typologies of serious games. This is a topic little explored by the authors. Is it the profile of the students that should justify this choice or type of learning outcomes?

Revision made: We added an explanation about this essential issue in the begging of the conclusion section (lines 868-889).

We remain at your disposal for any further comments and suggestions!

The authors

24 May 2022

Round 2

Reviewer 1 Report

Thanks for the revised version

Author Response

Comments to the reviewer: once again, thank you very much for the first and very detailed review. Regarding the minor changes required for the revised version, we did the following:

  1. In order to improve the English language and style, we did a detailed spell checking of all the text of the article and corrected all found errors and wordy/complex sentences.
  2. We revised the presentation of the empirical research and added some clarifications to present them in a better and more precise way. See section 4.2, especially lines 594-603 and 626-631.

Reviewer 3 Report

I have only a minor improvement suggestion:

- This sentence needs to be better justified and referenced in the literature:

“Didactic games increase learners’ awareness of the school subjects and crucial issues like healthcare, preserving the culture and historical heritage, climate changes, social causes, etc.”

Author Response

Comments to the reviewer: once again, thank you very much for the first and very detailed review. Regarding the minor changes required for the revised version, we did the following:

  1. In order to improve the English language and style, we did a detailed spell checking of all the text of the article and corrected all found errors and wordy/complex sentences.
  1. We remastered the sentence in question and provided a better justification accompanied by three new literature references (lines 63-66).